# A comprehensive economic assessment of the burden of obesity in Kuwait

Mouaddh Abdulmalik Nagi[1]*, Hanan Ahmed[1], Mohammad Almari[2,3], Ziyad S. Almalki[4], Yasir Mohammed Zaroug Elradi[5]

1 Department of Pharmacy, Faculty of Medical Sciences, Aljanad University for Science and Technology, Taiz, Yemen, 2 Department of Health Policy and Management, College of Public Health, Kuwait University, Kuwait City, Kuwait, 3 Department of Health Services Research and Policy, London School of Hygiene and Tropical Medicine, University of London, London, United Kingdom, 4 Department of Clinical Pharmacy, Prince Sattam Bin Abdulaziz University, Riyadh, Al-Kharj, Saudi Arabia, 5 Family Medicine Department, Dhaman Co, Kuwait City, Kuwait

* muadh.ye@gmail.com

## Abstract

### Background

Obesity is a complex public health issue that has risen to epidemic proportions globally. The aim of this study was to estimate the economic costs associated with obesity from governmental and societal perspectives in the State of Kuwait in 2024.

### Methods

A disease-specific prevalence-based cost-of-illness framework was applied. Key parameters include prevalence of obesity and its related comorbidities, relative risks, healthcare resources, income rate and growth, and lost workdays for patients and relatives, all were derived from literature. The outcomes measured were total healthcare costs, societal costs, and cost per patient, all reported in 2024 Kuwaiti Dinar (KWD), United States dollar ($), and Purchasing Power Parity (PPP).

### Results

In 2024, there were approximately 2 million morbidity cases and 961 mortality cases directly attributed to obesity in Kuwait. The economic burden of obesity was estimated at KWD 4.3 billion ($ 14 billion; PPP$ 21.6 billion) from societal perspective and KWD 3.9 billion ($12.8 billion, PPP 19.7 billion) from governmental perspective, almost 1.3 times healthcare budget. The mean annual societal cost per patient was KWD 1,737 ($5,676; PPP 8,729)—17.6% of 2024 Kuwaiti gross domestic product per capita. Furthermore, the mean direct medical cost per patient was KWD 1,586 ($5,184 or PPP 7,972); comprised 3 times the healthcare expenditure per capita.

**Data availability statement:** All relevant data are within the paper and its Supporting information files.

**Funding:** The author(s) received no specific funding for this work.

**Competing interests:** The authors have declared that no competing interests exist.

## Conclusion

Obesity and its comorbidities impose a far greater health and economic burden on Kuwait's healthcare system and national productivity than previously recognized. This study calls for a paradigm shift toward early prevention, culturally tailored strategies, and comprehensive disease management. Expanding access to emerging innovative treatments and leveraging technology-driven tools are essential to support sustained weight management and reduce long-term impacts.

## Introduction

Obesity represents a complex public health challenge that has escalated to epidemic levels worldwide [1], posing significant risks to morbidity and mortality [2]. It is clinically defined by a body mass index (BMI) of 30 kg/m² or higher, a threshold widely used to identify individuals at increased risk for obesity-related complications [2]. The consequences of obesity extend beyond individual health, profoundly impacting public health systems and national economies [3,4]. Besides the escalating healthcare expenditures, obesity imposes a substantial economic burden on societies by contributing to increased absenteeism, early retirement, disability, premature mortality, and diminished productivity among relatives [3,4]. In 2019, the combined burden of obesity and overweight was estimated to account for approximately 2.2% of the global gross domestic product (GDP), with projections indicating a rise to 3.3% by 2060 if current trends persist [3]. Resources currently lost to obesity-related health and productivity burdens could be reallocated to improve economic development, thereby enhancing societal well-being and economic resilience. This substantial economic loss highlights the significant potential gains achievable through effective obesity prevention and management strategies [5].

The Middle East and North Africa (MENA) region has witnessed one of the highest prevalence of overweight and obesity globally, with approximately 77.7% of adults aged 25 years and older affected in 2021. Moreover, the rate of increase in obesity within the MENA region has outpaced many other parts of the world, showing a 39.4% relative rise in obesity prevalence between 1990 and 2021 [1]. Particularly, the Gulf Cooperation Council (GCC) countries are experiencing a rapidly escalating prevalence of obesity, with prevalence rates among the highest worldwide [1,6]. This alarming trend is primarily driven by rapid urbanization, infrastructural development, and technological advancements that have fundamentally altered lifestyles. Key contributors include a marked decline in physical activity and shifts toward energy-dense and nutrient-poor diets [7]. Over recent decades, GCC countries—particularly Kuwait—have witnessed rapid proliferation of Western fast-food chains, expansion of 24/7 online food delivery services, rise in sugar-sweetened beverage consumption (2.5 liters per person per week), increase intake of sugary snacks, larger portion sizes, and late-night or irregular eating habits outside the home [8–10].

Within the GCC countries, Kuwait notably stands out for its high adult obesity prevalence. According to the World Health Organization (WHO), the prevalence of obesity

among adults in Kuwait was estimated at 41.4% (33.3% of males and 45.6% of females) in 2016 [11]. A more recent global pooled analysis of the obesity trends from 1990 to 2022 showed that 38.9% of males and 49.9% of females had obesity in Kuwait [12]. Additionally, another study analyzed the results of a national survey conducted between 2011 and 2014 reported an obesity prevalence of 42.1% among adults aged 18–82 years, with rates of 38.5% in men and 47.9% in women [13]. Despite these differences in prevalence estimates, all studies highlight a consistently high and concerning rate of obesity in Kuwait, positioning the country among the highest worldwide [10,12].

Even so, the economic burden of obesity in Kuwait remains incompletely understood. For instance, one global study estimated direct and indirect costs attributable to overweight and obesity in Kuwait and other countries from 2019 to 2060, considering 28 obesity-related diseases. Drawing on epidemiological and cost data from published studies and global datasets, it estimated Kuwait's 2019 burden at US$2.3 billion—1.7% of the country's GDP—with projections rising to US$33.3 billion (9.42% of the country's GDP) by 2060 [3]. That year, costs comprised direct medical (US$638 million), direct non-medical (US$360 million), and indirect (US$1,660 million). Relative to other countries, Kuwait's obesity-related economic burden ranks among the highest, exceeding many high-income countries (e.g., 1.30% of the national GDP in Korea, 1.53% in Qatar, and 0.96% in Singapore) [3]. Moreover, Kuwait also ranks among the highest countries in terms of mortality attributable to high BMI globally. In 2023, high BMI accounted for 21.77% of all deaths in Kuwait, compared with 8.06% in high-income countries and 6.12% worldwide [14]. Complementing these findings, a retrospective micro-costing study from the public payer perspective examined ten key obesity-related comorbidities, collecting healthcare resource utilization data through surveys of public sector clinicians and cost data from the Ministry of Health. This study reported 2022 annual direct healthcare costs per patient with a single obesity-related comorbidity ranging from US$5,682 to US$13,666 [15].

Despite these insights, both studies have notable limitations. For example, neither included costs of treating obesity itself. Furthermore, the global study excluded vital components such as direct non-medical costs (e.g., equipment, home renovation, and housekeeping costs), out-of-pocket expenditures (OOPEs) related to treatment, relatives' productivity losses, and informal care costs [3]. The micro-costing analysis focused solely on healthcare costs, omitting direct non-medical and indirect costs like absenteeism, early retirement, relatives' burden, and mortality costs [15]. Given that these elements constitute a significant portion of obesity' global economic burden, their absence leaves a substantial gap in comprehensively understanding the full societal and economic impact of obesity in Kuwait.

In this context, cost-of-illness (COI) studies offer a comprehensive approach, representing a key form of health economic analysis that quantifies the economic burden of diseases or risk factors like obesity on individuals and society. These analyses delineate direct medical costs (e.g., hospitalizations, pharmaceuticals), direct non-medical costs (e.g., transportation, accommodation), and indirect costs (e.g., productivity losses from morbidity and premature mortality), yielding reliable country-specific estimates vital for evaluating preventive or control interventions [16]. Ultimately, by informing resource allocation, prioritizing policies, raising awareness of obesity's societal impact, and justifying targeted budget measures—particularly in regions where such expenditures rival major health threats—COI studies underpin evidence-based decision-making and strategies extending beyond clinical outcomes [17].

The aim of this study is to estimate the economic costs associated with obesity, including direct healthcare costs, direct non-medical costs, productivity losses, and other indirect costs in the State of Kuwait during the year 2024. Furthermore, this study aims to identify the main drivers of the economic burden of obesity in Kuwait.

## Methods

### Study population and design

The study population included all individuals aged 20 and older living with obesity in Kuwait in 2024. The current COI study estimated the direct and indirect costs associated with obesity in Kuwait for the year 2024. Using a disease-specific, prevalence-based approach, the analysis was conducted from both societal and healthcare system perspectives. COI

studies assess the economic burden a disease or risk factor places on healthcare system and society by quantifying the resources used and productivity lost due to the condition, highlighting potential economic gains if the condition were eradicated [4,18]. These studies are vital for informing policymakers to set healthcare priorities and allocate limited resources efficiently. COI analyses typically consider both direct costs—such as healthcare services, diagnostics, medications—and indirect costs, like lost productivity and non-medical costs (e.g., transportation, meals) [18]. The prevalence-based method gives an overview of the current economic impact, often combined with methods like the obesity attributable fraction (OAF) to identify disease-specific costs [19] and the human capital method (HCM) to estimate productivity losses [4,20]. Together, these methods provide a comprehensive view of the economic implications of diseases, supporting healthcare planning and interventions. This framework was applied in studies assessing obesity-related costs among adults to guide policy and budget decisions aimed at mitigating obesity impacts [3,4]. Importantly, to ensure methodological rigor, reporting quality, and international comparability, this study followed the consensus-based checklist for critical appraisal of COI studies developed by Schnitzler et al. (2023) [21]. This checklist comprises 17 main questions (plus sub questions) across three domains: study characteristics, methodology and cost analysis, and results and reporting. Notably, it has been adopted in several published COI studies [22,23].

## Obesity-associated diseases

This study included obesity and the most common chronic diseases related to obesity among the adult population (32 diseases), as reported by recent large-scale epidemiological studies [2], Table 1 and S1 Table. These diseases are grouped into nine main categories; 1) type II diabetes mellitus (T2DM); 2) cardiovascular diseases (CVDs) including ischemic heart diseases (IHDs), stroke, atrial fibrillation and flutter, and hypertensive heart diseases; 3) chronic kidney diseases (CKDs); 4) cancers covering postmenopausal breast cancer, uterine/endometrial, ovary, colorectal, esophagus adenocarcinoma, gastric cardia, liver, gallbladder, pancreas, prostate, kidney, thyroid, lymphoma, multiple myeloma, meningioma, and leukemia; 5) musculoskeletal disorders comprising osteoarthritis, back pain, and gout; 6) gastrointestinal disorders including gastro-esophageal reflux disease (GERD), nonalcoholic fatty liver (NAFL), and gallbladder disease; 7) asthma; and 8) mental and neurological disorders comprising depressive disorders and alzheimer's disease and other dementias; 9) and poly-cystic ovary syndrome (PCOS).

## Obesity attributable fraction (OAF)

To quantify the burden of obesity, it is essential to estimate the OAF, which measures the proportion of disease cases and related costs that can be attributed to obesity by estimating the fraction of cases or deaths that would be prevented if obesity were eliminated, thus isolating the economic impact directly linked to obesity [34]. The OAF is estimated using the following formula [4,19]:

$$OAFi\,(\%) = \frac{P\,(RR_i - 1)}{1 + P\,(RR_i - 1)} \times 100$$

where $RRi$ the relative risk of obesity for disease $i$ (separately estimated for morbidity and mortality), and $P$ is the prevalence of obesity among the adult population (≥ 20 years) in Kuwait. $RR$s for all obesity-associated diseases included in this analysis were sourced from robust meta-analyses and cohort studies [24–33,35,36] (Table 1).

On the other hand, the prevalence of obesity in Kuwait was obtained from WHO 2016 report, which indicated that 33.3% of adult males and 45.6% of adult females were affected [11]. The year 2016 was specifically chosen to account for the latency period between obesity exposure and the development of obesity-related outcomes, including morbidity and mortality. Since most obesity-related conditions are chronic, there is an inherent time lag between the onset of obesity and the manifestation of disease or death. Consequently, morbidity and mortality data from 2024 reflect obesity exposures that

**Table 1. Relative risks of obesity-associated diseases.**

| No | Disease | Morbidity relative risk | | Ref | Mortality relative risk | | Ref |
|---|---|---|---|---|---|---|---|
| | | Male | Female | | Male | Female | |
| 1 | Esophagus cancer | 1.21 | 1.20 | [24] | 1.39 | 1.35 | [25] |
| 2 | Gastric cancer | 1.8 | 1.8 | [26] | 1.04 | 1.04 | [27] |
| 3 | Colorectal cancer | 1.95 | 1.66 | [24] | 1.29 | 1.05 | [28] |
| 4 | Liver cancer | 1.8 | 1.8 | [26] | 1.47 | 1.47 | [28] |
| 5 | Pancreas cancer | 2.29 | 1.60 | [24] | 1.07 | 1.09 | [25] |
| 6 | Breast cancer | NA | 1.13 | [24] | NA | 1.15 | [28] |
| 7 | Uterine/Endometrial cancer | NA | 7.1 | [26] | NA | 1.77 | [27] |
| 8 | Ovary cancer | NA | 1.28 | [24] | NA | 2.62 | [27] |
| 9 | Prostate cancer | 1.05 | NA | [24] | 1.45 | NA | [27] |
| 10 | Kidney cancer | 1.82 | 2.64 | [24] | 1.59 | 1.59 | [27] |
| 11 | Meningioma | 1.50 | 1.50 | [26] | 1.50 | 1.50 | [29] |
| 12 | Gallbladder cancer | 1.3 | 1.3 | [26] | 1.16 | 1.34 | [25] |
| 13 | Thyroid cancer | 1.10 | 1.10 | [26] | 1.22 | 1.14 | [25] |
| 14 | Non-Hodgkin Lymphoma | 1.40 | 1.34 | [30] | 1.09 | 1.13 | [25] |
| 15 | Multiple myeloma | 1.50 | 1.50 | [26] | 1.20 | 1.20 | [27] |
| 16 | Leukemia | 1.09 | 1.13 | [25] | 1.66 | 1.66 | [27] |
| 17 | Type II diabetes mellitus | 6.47 | 12.41 | [24] | 2.16 | 2.16 | [28] |
| 18 | Hypertensive heart disease | 1.84 | 2.42 | [24] | 2.03 | 2.03 | [28] |
| 19 | Ischemic heart diseases | 1.72 | 3.10 | [24] | 1.39 | 1.39 | [28] |
| 20 | Stroke | 1.51 | 1.49 | [24] | 1.39 | 1.39 | [28] |
| 21 | Atrial fibrillation and flutter | 1.34 | 1.35 | [25] | 1.34 | 1.35 | [25] |
| 22 | Chronic kidney disease | 1.73 | 1.73 | [25] | 1.73 | 1.73 | [25] |
| 23 | Asthma | 1.43 | 1.78 | [24] | 1.41 | 1.40 | [25] |
| 24 | Gastro-esophageal reflux disease | 2.00 | 2.00 | [31] | NA | NA | – |
| 25 | Nonalcoholic fatty liver | 3.53 | 3.53 | [32] | 1.82 | 1.82 | [28] |
| 26 | Gallbladder disease | 1.43 | 2.32 | [24] | 1.82 | 1.82 | [28] |
| 27 | Depressive disorders | 1.31 | 1.67 | [25] | NA | NA | – |
| 28 | Alzheimer's disease and other dementias | 1.22 | 1.21 | [25] | 1.22 | 1.21 | [25] |
| 29 | Osteoarthritis | 4.20 | 1.96 | [24] | NA | NA | – |
| 30 | Back pain | 2.81 | 2.81 | [24] | NA | NA | – |
| 31 | Gout | 1.63 | 1.49 | [25] | NA | NA | – |
| 32 | Polycystic ovarian syndrome | NA | 2.77 | [33] | NA | NA | – |

NA: Not available/applicable.

occurred several years earlier, rather than more recent ones. In this context, an eight-year lag was selected as a practical compromise balancing data availability and methodological considerations. This approach is consistent with global trends; a recent systematic review found that over two-thirds (68.4%) of COI studies on obesity published between 2016 and 2022 used latency periods shorter than five years, while only about one-third applied longer lag times exceeding six years [4].

## Cost estimation

This analysis was conducted from both the healthcare system and societal perspectives. From the healthcare system perspective, the focus was on costs directly incurred by the healthcare sector in providing obesity-related services, including

primary care, diagnostics, medications, medical supplies, surgeries, and more. In contrast, the societal perspective encompassed all costs regardless of who bears them. This broader view included OOPE for treatment, direct non-medical costs such as transportation, meals, accommodation, equipment (e.g., walking aid, bed, chair, supporting stick, wheelchair, home exercise equipment, etc.), home modifications, and housekeeping services, as well as indirect costs. Indirect costs accounted for productivity losses due to absenteeism, reduced work output of relatives and housekeepers, and premature mortality. By incorporating these varied cost components, the societal perspective provides a comprehensive estimate of the economic burden of obesity [3,4].

**Direct costs.** The direct costs of obesity comprise healthcare costs for treating obesity itself, healthcare costs related to treating obesity-associated comorbidities, OOPE incurred by individuals with obesity, and non-medical costs linked with obesity. To estimate the direct medical costs of obesity treatment, the population of individuals living with obesity in Kuwait in 2024 was first determined based on projections from the World Obesity Atlas for 2022–2030 (S1 Table). The atlas estimated that 2,471,876 individuals in Kuwait were living with obesity in 2024, comprising 1,358,873 men (55%) and 1,113,003 women (45%) [37]. For estimating the direct medical costs of obesity, excluding metabolic bariatric surgery (MBS), a treatment scenario from the literature was applied wherein only 2% of individuals living with obesity (totaling 49,438; 27,177 males and 22,260 females) receive formal obesity treatment [38,39], which in fact validated by real-world data from neighboring countries [17]. Among these treated patients, 18,391 (37.2%) were on Wegovy® (Semaglutide), 15,474 (31.3%) took Saxenda® (Liraglutide), 13,744 (27.8%) on Mounjaro® (Tirzepatide), 1,038 (2.1%) took two of these medications, and 791 (1.6%) on all three medications [40]. To estimate the treatment costs, the number of individuals receiving treatment was multiplied by the annual cost per patient for each corresponding medication. This approach reflects current treatment patterns in the country.

Extensionally, the cost of MBS was calculated separately by multiplying the number of surgeries by the annual per-person cost, which encompasses both the surgical procedure and the management of potential complications and any necessary re-surgeries. Despite the great demand for MBS noticed lately in Kuwait and neighboring countries [41,42], a compound annual growth rate of 12.42% was applied on the number of MBS since 2019 [43]. Consequently, 5,296 MBSs were factored into our cost estimations.

In contrary, this study extracted the number of cases of the 32 obesity-associated morbidities considered in the analysis from the latest Global Burden of Disease (GBD) study [14] (S1 Table). Subsequently, OAF was multiplied by number of morbidity cases of each disease ($OAF \times number\ of\ people\ affected\ by\ the\ disease$) to obtain the number of morbidity cases of that disease attributable to obesity. Then, the number of morbidity cases of each disease attributable to obesity was multiplied by the cost of treating that disease ($attributable\ cases\ of\ a\ disease \times the\ annual\ per-person\ cost\ of\ that\ disease$).

In the absence of primary cost data specific to Kuwait, this study adopted a hierarchical approach using data from high-income countries (HICs), a methodologically accepted approach in economic evaluations when context-specific data are unavailable but reliable estimates are necessary to inform decision-making [44,45]. For example, 62.5% of health economic evaluations published between 2000 and 2022 in GCC countries transferred direct healthcare costs from other jurisdictions [46]. The rationale for selecting HICs stems from their comparable healthcare system characteristics with Kuwait's, which enhances the relevance and applicability of their economic data. HICs commonly feature advanced healthcare infrastructure, established financing mechanisms, equitable access to services, efficient administrative systems, and high capacity for universal health coverage, all of which influence healthcare costs and utilization patterns in ways more aligned with Kuwait's healthcare environment than low- and middle-income countries (LMICs) [47,48].

The hierarchy for cost data collection prioritized Kuwaiti data, then expanded to other GCC countries—Bahrain, Oman, Qatar, Saudi Arabia, and the United Arab Emirates—due to their socioeconomic and cultural similarities, comparable disease profiles (especially for obesity-related conditions), and similar healthcare delivery models and resource availability. If data remained unavailable from these regional sources, information from other HICs was used as a last resort (S2 Table). To ensure data standardization and comparability, costs of obesity-related diseases reported in original

studies were first converted to the local currency of the source country—when not originally reported in local currency—at the year of estimation. These figures were then inflated to 2024 values using the average Consumer Price Index (CPI) reported by the International Monetary Fund (IMF) World Economic Outlook Database of October 2024 [49]. Subsequently, these costs were adjusted to 2024 international dollar/Purchasing Power Parity (PPP) values using implied PPP conversion rates from the same IMF database [49]. Finally, the amounts were converted into Kuwaiti Dinar (KWD) and United States dollars (US$) based on exchange rates (1 KWD ≈ 5.03 PPP and 3.27 US$) as of July 1st, 2024, provided by Oanda Corporation [50].

Furthermore, to estimate the total OOPE and direct non-medical costs associated with obesity, the previously determined number of individuals living with obesity in Kuwait in 2024 was multiplied by the respective annual per-person costs. These annual per-person OOPE for medical treatment and direct non-medical costs were sourced from a recent study conducted in Saudi Arabia [17] and underwent the same inflation and adjustment steps done for direct medical costs (S2 Table). In the meantime, OOPE encompassed medical costs not covered by health insurance. Direct non-medical costs included costs for meals, transportation, and accommodation during hospital visits or stays, as well as housekeeping, home renovations (e.g., structural modifications), and adaptive equipment (e.g., walking aids, wheelchairs) required by patients to manage the condition [17].

**Indirect costs.** Indirect costs represent the foregone opportunities of earning that encompass lost productivity due to absenteeism by patients and their relatives or any other companions (while seeking or waiting for obesity medical care), as well as premature mortality. The cost of absenteeism was quantified by multiplying the average number of workdays lost by people living with obesity and their companions by daily wages, proxied by 2024 Kuwait GDP per capita. The annual average number of workdays lost as a result of obesity and its comorbidities was derived from recent study conducted in Saudi Arabia [17] while 2024 Kuwait GDP per capita (US$32,214) was obtained from the World Bank database [51] (S1 Table). On the other side, premature mortality costs were estimated using the human capital approach, which values lost productivity as the present value of lifetime earnings (PVLE) from the time of death until expected life expectancy. For context, this calculation incorporated several data elements such as personal income (2024 Kuwaiti GDP per capita; US$32,214), life expectancy, country's rate of economic growth [51], and a 3% discount rate per the WHO guidelines [52]:

$$Present\ value\ of\ lifetime\ earnings\ =\ \frac{future\ value}{(1\ +\ discount\ interest\ rate)^{cycle}}$$

Mortality data for 26 obesity-related comorbidities were sourced from the latest GBD study [14], assumed constant for 2024 due to lack of more recent data. It should be noted that six diseases (i.e., GERD, depressive disorders, osteoarthritis, back pain, gout, and PCOS) were not included in the mortality cost estimation as they lack mortality data. Likewise, data on life expectancy were obtained from the analysis of population forecasting scenarios for the GBD study [53]. Subsequently, OAF was multiplied by total number of deaths for each disease ($OAF\ \times\ number\ of\ people\ died\ due\ to\ the\ disease\ in\ 2024$) to obtain the number of deaths of that disease attributable to obesity. The years lost were calculated as the difference between age at death and life expectancy. Then, the number of attributable deaths was multiplied by the PVLE to estimate productivity losses from premature mortality ($attributable\ deaths \times PVLE$). This method holistically integrated mortality, economic, and demographic data to quantify indirect costs attributable to obesity.

Direct non-medical costs and productivity losses from absenteeism were attributed to obesity as the primary risk factor, not to individual comorbidities, to avoid double counting. This approach, consistent with the attributable fraction framework, consolidates costs under the underlying causal condition rather than dispersing them among associated diseases. Assigning these costs separately to each obesity-related disease risks overlapping estimates, as non-medical and

productivity losses are often shared across multiple conditions and cannot be reliably separated in cases of multimorbidity. As such, non-medical and absenteeism costs were aggregated at the obesity level. This method aligns with established COI guidelines that recommend avoiding overlapping cost attribution in multi-condition analyses [4,18,20].

### Sensitivity analysis

To evaluate the impact of parameter uncertainty, sensitivity analyses were conducted on key input variables, including the prevalence of obesity, the prevalence of obesity-associated comorbidities, and the costs of treating obesity-related diseases. These analyses were designed to provide a robust framework to quantify the influence of key uncertainties on economic burden estimates and support informed policy decision-making. The first scenario simulated a 20% reduction in the prevalence of obesity and its comorbidities. This hypothetical yet attainable improvement illustrates the potential economic benefits arising from targeted reductions in obesity prevalence achievable through public health interventions such as lifestyle modification programs, awareness campaigns, and behavioral regulations. Similar reductions have been documented in countries implementing comprehensive prevention strategies, supporting the use of this assumption as a practical baseline for policy evaluation [54,55]. A second scenario examined reductions of 30% and 10% in the annual per-patient treatment costs for obesity-related diseases, addressing uncertainties in nationwide treatment expenditures and illustrating the potential impact of improved healthcare system efficiency [56]. Finally, given the recent upward trend in obesity prevalence observed in Kuwait, sensitivity analyses incorporated the rates reported by Phelps et al. (2024)—38.9% in males and 49.9% in female [12] to model the potential impact of continued increases if no intervention is implemented.

## Results

### Health burden

According to OAF estimates, obesity was responsible for 2,067,311 cases, representing 42% of total morbidity cases (4,903,456) caused by the diseases included in this study among adults aged 20 years or older in Kuwait. Specifically, obesity accounted for 72%, 49%, and 45% of the respective disease burdens of T2DM, NAFL, and PCOS. The highest number of morbidity cases was reported for NAFL (860,612 cases), followed by T2DM (458,859 cases) and GERD (186,448 cases). Conversely, diseases such as liver cancer, esophagus adenocarcinoma, and gallbladder and biliary cancers, with only 2, 4, and 4 cases respectively, were among the lowest in terms of attributable morbidity. Likewise, among 6,636 total deaths from the included diseases, obesity contributed to 961 premature deaths, equating to 14% of deaths. IHD resulting in 347 deaths, followed by T2DM (167 deaths) and hypertensive heart disease (103 deaths). On the opposite end, thyroid cancer, liver cancer, and gallbladder biliary cancers contributed minimally, with just 0–2 reported attributable deaths each (Table 2). Additionally, the findings revealed that premature mortality attributable to obesity resulted in 20,062 years of potential life lost (YPLL) in 2024, meaning an average of 21 potential years of life per obesity-related death. The most significant contributors to YPLL were IHD (8,007 years), T2DM (3,066 years) and stroke (1,802 years). In comparison, thyroid cancer (20 years), gastric cancer (23 years), and liver cancer (23 years) accounted for the fewest YPLL. Collectively, CVDs represent the most severe impact, with 11,576 YPLL reported (58% of the total) (Table 2).

### Economic burden

The total economic burden of obesity in 2024 was substantial, amounting to approximately KWD 4.29 billion ($14.03 billion, PPP 21.58 billion) in both public and private healthcare facilities. This figure includes KWD 3.92 billion ($12.82 billion, PPP 19.71 billion) in direct medical costs, KWD 38 million ($124 million, PPP 191 million) in direct non-medical costs, KWD 225.45 million ($736.7 million, PPP 1.13 billion) in indirect morbidity costs (absenteeism costs), and KWD 180.75 million ($355.36 million, PPP 546.46 million) in mortality costs. Of the direct medical costs, only 0.79%

Table 2. OAF, morbidity cases, mortality cases, and YPLL attributable to obesity in Kuwait.

| Disease | Morbidity | | | | Mortality | | | | YPLL | |
| --- | --- | --- | --- | --- | --- | --- | --- | --- | --- | --- |
| | OAF (%) | | Attributable cases | | OAF (%) | | Attributable deaths | | | |
| | Male | Female | Male | Female | Male | Female | Male | Female | Male | Female |
| Esophagus cancer | 7 | 8 | 3 | 2 | 12 | 14 | 2 | 1 | 44 | 21 |
| Gastric cancer | 21 | 27 | 33 | 7 | 1 | 2 | 1 | 0 | 14 | 9 |
| Colorectal cancer | 24 | 23 | 573 | 377 | 9 | 2 | 14 | 12 | 320 | 44 |
| Liver cancer | 21 | 27 | 1 | 1 | 14 | 18 | 1 | 0 | 11 | 12 |
| Pancreas cancer | 30 | 21 | 24 | 7 | 2 | 4 | 2 | 1 | 44 | 33 |
| Breast cancer | NA | 6 | NA | 652 | NA | 6 | NA | 10 | NA | 367 |
| Uterine/Endometrial cancer | NA | 74 | NA | 2,076 | NA | 26 | NA | 8 | NA | 231 |
| Ovary cancer | NA | 11 | NA | 51 | NA | 42 | NA | 16 | NA | 550 |
| Prostate cancer | 2 | NA | 87 | NA | 13 | NA | 14 | NA | 186 | NA |
| Kidney cancer | 21 | 43 | 116 | 172 | 16 | 21 | 5 | 2 | 120 | 44 |
| Meningioma | 14 | 19 | 18 | 6 | 14 | 19 | 6 | 3 | 173 | 107 |
| Gallbladder cancer | 0 | 19 | 0 | 4 | 5 | 14 | 1 | 1 | 18 | 34 |
| Thyroid cancer | 3 | 4 | 27 | 70 | 7 | 6 | 0 | 0 | 10 | 10 |
| Non-Hodgkin Lymphoma | 12 | 13 | 166 | 97 | 3 | 6 | 1 | 1 | 44 | 50 |
| Multiple myeloma | 14 | 19 | 10 | 9 | 6 | 8 | 1 | 1 | 21 | 20 |
| Leukemia | 3 | 6 | 9 | 6 | 18 | 23 | 10 | 5 | 286 | 195 |
| Type II diabetes mellitus | 65 | 84 | 241,288 | 217,572 | 28 | 35 | 102 | 64 | 1812 | 1254 |
| Hypertensive heart disease | 22 | 39 | 900 | 1,615 | 26 | 32 | 49 | 53 | 751 | 930 |
| Ischemic heart diseases | 19 | 49 | 29,776 | 37,075 | 11 | 15 | 281 | 66 | 6626 | 1381 |
| Stroke | 15 | 18 | 3,492 | 3,193 | 11 | 15 | 64 | 30 | 1187 | 615 |
| Atrial fibrillation and flutter | 10 | 14 | 648 | 481 | 10 | 14 | 4 | 3 | 41 | 45 |
| Chronic kidney disease | 20 | 25 | 42,632 | 43,371 | 20 | 25 | 43 | 38 | 792 | 880 |
| Asthma | 13 | 26 | 6,289 | 11,659 | 12 | 15 | 2 | 2 | 35 | 38 |
| Gastro-esophageal reflux disease | 25 | 31 | 80,611 | 105,837 | 0 | 0 | 0 | 0 | 0 | 0 |
| Nonalcoholic fatty liver | 46 | 54 | 441,749 | 418,862 | 21 | 27 | 4 | 2 | 80 | 52 |
| Gallbladder disease | 13 | 38 | 4,678 | 27,626 | 21 | 27 | 3 | 2 | 43 | 46 |
| Depressive disorders | 9 | 23 | 6,662 | 22,504 | 0 | 0 | 0 | 0 | 0 | 0 |
| Alzheimer's disease and other dementias | 7 | 9 | 574 | 681 | 7 | 9 | 17 | 20 | 188 | 249 |
| Osteoarthritis | 52 | 30 | 67,299 | 38,509 | 0 | 0 | 0 | 0 | 0 | 0 |
| Back pain | 38 | 45 | 66,495 | 101,649 | 0 | 0 | 0 | 0 | 0 | 0 |
| Gout | 17 | 18 | 3,502 | 1,020 | 0 | 0 | 0 | 0 | 0 | 0 |
| Polycystic ovarian syndrome | 0 | 45 | 0 | 34,456 | 0 | 0 | 0 | 0 | 0 | 0 |
| Total | 39 | 46 | 997,662 | 1,069,647 | 13 | 18 | 628 | 342 | 0 | 0 |

OAF: obesity attributable fraction; NA: Not available/applicable; YPLL: years of potential life lost.

(around KWD 31.10 million, $101.63 million, PPP 510.7 million) was related to the treatment of obesity itself while the rest was related to the associated comorbidities. When considering the total costs—including both direct and indirect costs—of each single disease, T2DM stood out as the highest contributor, incurring approximately KWD 1.23 billion ($4.03 billion, PPP 6.20 billion). This was followed closely by NAFL, with total costs nearing KWD 1.04 billion ($3.40 billion, PPP 5.22 billion), and GERD, with KWD 474.67 million ($1.55 billion, PPP 2.39 billion). In contrast, diseases such as liver cancer KWD 152,060 ($496,896, PPP 764,121), gallbladder and biliary cancers KWD 361,525 ($1.18 million, PPP 1.82 million),

and esophagus adenocarcinoma KWD 420,894 ($1.38 million, PPP 2.12 million) presented the lowest economic impact attributable to obesity (Table 3).

A comparison of cost components indicated that direct medical costs KWD 3.92 billion ($12.82 billion, PPP 19.71 billion) far account for the highest part of obesity burden in Kuwait (91.33%) while the share of direct non-medical costs represents 0.88%. Nonetheless, indirect costs of productivity loss due to absenteeism and premature mortality (KWD 334.20 million, $1.09 billion, PPP 1.68 billion) present a considerable component (7.78%) of the total economic burden in the country (Table 3). An analysis of the total costs across various disease groups reveals substantial variability in the economic burden. Notably, GIT diseases emerge as the most burdensome, with total costs amounting to KWD 1.55 billion ($5.08 billion, PPP 7.80 billion or 36% of the total economic burden). T2DM ranks second, incurring KWD 1.23 billion ($4.03 billion, PPP 6.20 billion or 29% of the total economic burden), driven predominantly by high direct medical costs. CVDs and CKD also impose considerable economic burdens, with total costs of KWD 393 million ($1.29 billion, PPP 1.98 billion or 9% of the total economic burden) and KWD 362 million ($1.18 billion, PPP 1.82 billion or 8% of the total economic burden), respectively. At the lower end of the spectrum, asthma and PCOS contribute KWD 10.06 million ($32.86 million, PPP 50.53 million) and KWD 23.97 million ($78.34 million, PPP 120.47 million) in total costs, respectively. Musculoskeletal disorders, despite a notable case burden (278,474 cases), result in a comparatively modest total cost of KWD 296.81 million ($970 million, PPP 1.49 billion). Obesity itself is particularly notable in this analysis; despite the absence of reported mortality costs, it generates KWD 294.49 million ($962.33 million, PPP 1.48 billion) in total costs through significant indirect morbidity costs. In summary, this comparative analysis reveals that GIT diseases and T2DM are the leading contributors to both direct and total healthcare costs, whereas CVDs have the most profound impact in terms of premature mortality and YPLL. While conditions such as asthma, PCOS, and neuro/mental disorders impose relatively lower economic burdens (S3 Table).

From a macroeconomic point of view, the economic burden of obesity was estimated at KWD 4.29 billion ($14.03 billion; PPP 21.58 billion) from societal perspective and KWD 3.92 billion ($12.82 billion; PPP 19.71 billion) from governmental perspective, equivalent to 1.3 times the Ministry of Health budget in 2024—KWD 3.04 billion ($10 billion; PPP 15.3 billion) [57]. The mean annual societal cost per patient was KWD 1,737 ($5,676; PPP 8,729)—17.62% of 2024 Kuwaiti GDP per capita. Furthermore, the mean direct medical cost per patient was KWD 1,586 ($5,184 or PPP 7,972); comprised 3 times the health expenditure per capita KWD 520 ($1,700, PPP 2,614) [58].

### Findings of sensitivity analysis

The sensitivity analysis indicated that the prevalence of obesity and its comorbidities, along with the cost of treating obesity-related diseases, have the greatest impact on the total societal cost of obesity. A 20% decline in the prevalence of obesity and its comorbidities resulted in a 28% reduction in the total societal cost. Similarly, a 30% and 10% decrease in the cost of treating obesity-related diseases would result in corresponding declines of 27% and 9% in the total societal cost of obesity. Furthermore, adopting the prevalence rates reported by Phelps et al. (2024) resulted in a 6% increase in the total societal cost compared to the inputs used in the base-case analysis (Table 4).

### Discussion

This study provides a comprehensive evaluation of the health and economic burden of obesity in Kuwait, revealing critical insights for public health policy and clinical practice. Key findings indicate that obesity accounted for 2.07 million cases of morbidity and 961 premature deaths, representing 42% and 14%, respectively, of the total disease burden. Additionally, obesity accounted for 20,062 years of potential life lost (YPLL). Economically, obesity imposes a heavy burden from societal perspective (KWD 4.3 billion, $14.03 billion, or PPP 21.58 billion) and healthcare system perspective (KWD 3.92 billion, $12.82 billion, PPP 19.71 billion). This is substantially higher than previous estimates, such as the 2019 global study that reported an economic burden of $2.3 billion [3] from the societal perspective. This sharp increase reflects both the rising

**Table 3. Costs of obesity and attributable diseases in Kuwait (Kuwaiti Dinar).**

| Disease | Direct medical costs | | Direct non-medical costs | | Absenteeism costs | | Indirect mortality costs | |
|---|---|---|---|---|---|---|---|---|
| | Male | Female | Male | Female | Male | Female | Male | Female |
| Esophagus adenocarcinoma | 43,088 | 28,963 | NA | NA | NA | NA | 242,636 | 106,207 |
| Gastric cancer | 539,391 | 111,950 | NA | NA | NA | NA | 75,110 | 42,822 |
| Colon and rectum cancers | 11,248,633 | 7,402,163 | NA | NA | NA | NA | 1,689,294 | 208,888 |
| Liver cancer | 16,060 | 15,618 | NA | NA | NA | NA | 59,034 | 61,348 |
| Pancreas cancer | 380,491 | 107,825 | NA | NA | NA | NA | 234,827 | 168,474 |
| Breast cancer | NA | 10,828,654 | NA | NA | NA | NA | NA | 1,583,176 |
| Uterine/Endometrial cancer | NA | 33,516,538 | NA | NA | NA | NA | NA | 1,094,723 |
| Ovary cancer | NA | 363,506 | NA | NA | NA | NA | NA | 2,425,528 |
| Prostate cancer | 1,225,883 | NA | NA | NA | NA | NA | 1,261,742 | NA |
| Kidney cancer | 1,876,513 | 2,769,581 | NA | NA | NA | NA | 623,925 | 205,542 |
| Meningioma (Brain, central nervous system) | 297,289 | 97,033 | NA | NA | NA | NA | 787,105 | 443,932 |
| Gallbladder cancer | 54,868 | 37,437 | NA | NA | NA | NA | 96,629 | 172,591 |
| Thyroid cancer | 437,277 | 1,124,813 | NA | NA | NA | NA | 52,864 | 47,873 |
| Non-Hodgkin lymphoma | 2,686,026 | 1,566,729 | NA | NA | NA | NA | 201,464 | 215,393 |
| Multiple myeloma | 168,085 | 146,018 | NA | NA | NA | NA | 111,779 | 94,700 |
| Leukemia | 141,814 | 97,184 | NA | NA | NA | NA | 1,296,290 | 798,270 |
| Type II Diabetes mellitus | 639,016,448 | 576,207,482 | NA | NA | NA | NA | 10,675,407 | 7,193,920 |
| Hypertensive heart disease | 556,605 | 998,797 | NA | NA | NA | NA | 4,657,563 | 5,575,927 |
| Ischaemic heart disease | 112,561,722 | 140,155,258 | NA | NA | NA | NA | 34,223,607 | 7,521,227 |
| Stroke | 35,885,259 | 32,815,067 | NA | NA | NA | NA | 6,685,312 | 3,387,223 |
| Atrial fibrillation and flutter | 4,784,438 | 3,546,980 | NA | NA | NA | NA | 291,426 | 314,337 |
| Chronic kidney disease | 174,966,361 | 177,996,587 | NA | NA | NA | NA | 4,447,097 | 4,557,391 |
| Asthma | 3,380,656 | 6,266,835 | NA | NA | NA | NA | 200,527 | 207,253 |
| Gastro-esophageal reflux disease | 205,222,573 | 269,444,659 | NA | NA | NA | NA | NA | NA |
| Nonalcoholic fatty liver | 532,951,846 | 505,339,443 | NA | NA | NA | NA | 446,339 | 287,531 |
| Gallbladder diseases | 5,643,223 | 33,329,356 | NA | NA | NA | NA | 257,317 | 256,628 |
| Depressive disorders | 6,878,445 | 23,236,820 | NA | NA | NA | NA | NA | NA |
| Alzheimer's disease and other dementias | 535,808 | 635,954 | NA | NA | NA | NA | 1,369,415 | 1,788,285 |
| Osteoarthritis | 70,485,975 | 40,332,546 | NA | NA | NA | NA | NA | NA |
| Back pain | 69,644,088 | 106,463,511 | NA | NA | NA | NA | NA | NA |
| Gout | 7,656,244 | 2,230,188 | NA | NA | NA | NA | NA | NA |
| Polycystic ovarian syndrome | NA | 23,972,761 | NA | NA | NA | NA | NA | NA |
| Obesity | 11,841,844 | 19,258,565 | 20,860,800 | 17,086,323 | 123,935,094 | 101,510,734 | NA | NA |
| **Total** | **1,901,126,951** | **2,020,444,823** | **20,860,800** | **17,086,323** | **123,935,094** | **101,510,734** | **69,986,707** | **38,759,189** |

NA: Not applicable.

obesity prevalence and the improved inclusion of broader cost domains in the present analysis. In contrast to previous Kuwaiti estimates that focused solely on healthcare costs [15], the current study offers a broader societal lens by incorporating productivity losses (indirect costs) due to absenteeism and premature mortality, amounting to KWD 334.2 million ($1.09 billion, PPP 1.68 billion). This expands understanding of the burden beyond clinical treatment costs. A similar

**Table 4. Findings of sensitivity analysis.**

| Scenario | Total Costs (KWD billion) | Change from the base case (%) |
|---|---|---|
| Base case | 4.29 | NA |
| Reducing the prevalence of obesity and its comorbidities by 20% | 3.10 | −28 |
| Reducing the cost of treating obesity-associated diseases by 30% | 3.13 | −27 |
| Reducing the cost of treating obesity-associated diseases by 10% | 3.90 | −9 |
| Prevalence reported by Phelps et al. (2024) | 4.56 | 6 |

KWD: Kuwaiti Dinar; NA: Not applicable.

approach was recently taken in Saudi Arabia, where indirect costs were found to comprise around 12% of total obesity-related burdens [17], nearly matching the 7.78% found in our study.

When compared to recent COI studies from high-income countries like Portugal [59], Saudi Arabia [17], and Italy [60], several similarities emerge, such as the high contribution of T2DM and CVDs to total costs, the consistent dominance of direct medical costs over the economic burden, and the relatively lower proportion of obesity treatment costs versus comorbidity costs. For example, direct medical costs represented approximately 85% of the total obesity-related expenditures and the proportion of obesity treatment costs accounted for ~1% of the total obesity-related expenditures in Portugal [59]. Likewise, a recent COI from Saudi Arabia reported that direct medical costs represented 94% of the total cost and the obesity treatment costs accounted for ~2% [17]. Similarly, Switzerland's COI showed high direct cost proportions (97%), with T2DM and CVDs as leading cost drivers, while only 3% of the total costs were due to the combined direct management of overweight and obesity [61]. In contrast, earlier estimates reported direct medical costs comprising 43% of the total cost in the USA in 2019 [3] and 23.4% in 2023 [5]. In Kuwait, the findings of the present analysis indicated that the majority of the obesity burden come from managing obesity-associated comorbidities (90.6%) while obesity treatment costs accounted for only 0.72%.

The disproportionate economic burden incurred by obesity-related comorbidities (KWD 4.3 billion, $14.03 billion, or PPP 21.58 billion) compared to the relatively low direct costs of treating obesity itself (KWD 3.92 billion, $12.82 billion, PPP 19.71 billion), illustrates the low rates of obesity diagnosis and treatment at early stages which may lead to more complex, costly healthcare needs and poorer health outcomes and consequently to such a significant economic burden. In Kuwait and neighboring countries, obesity is frequently undiagnosed despite its high prevalence [62]. This is partly due to patients' limited recognition of obesity as a medical condition and consequently low healthcare-seeking behavior, as well as healthcare providers' reluctance to initiate weight-lowering therapies in routine clinical practice [63–65]. Contributing to poor diagnosis are insufficient routine screening protocols and the absence of dedicated prevention and treatment centers. Even when diagnosis occurs, evidence suggests that patient acceptance and adherence to obesity treatment are commonly low, often ranging between 2% and 10% [38,39,63,64,66]. This discrepancy indicates the underutilization of dedicated obesity treatment modalities, a pattern mirrored in other HICs [39,63,64,66] and reflects a healthcare system focused more heavily on the treatment of obesity-related complications rather than prevention or early intervention, which highlights a significant missed opportunity. Early targeted interventions could prevent progression to costly chronic diseases such as T2DM, NAFL, GERD, CKDs, and IHDs, which together comprise a cost over KWD 3.40 billion ($ 11 billion, PPP 17.10 billion) equivalent to 79.3% of the total economic burden. These findings call for a paradigm shift toward preventive care strategies aimed at addressing obesity as a root cause rather than treating its downstream consequences. In the context, while only 2% of individuals with obesity receive formal obesity treatment, this analysis assumed that all individuals with obesity-associated diseases received the needed care. This assumption likely contributes to the high estimated costs associated with obesity-related comorbidities.

On the other side, despite the minimal direct medical costs associated with obesity itself, the substantial indirect costs (KWD 225.45 million, $736.70 million, and PPP 1.133 billion) and associated non-medical costs (KWD 38 million, $124 million, and PPP 190.69 million) highlight a significant yet less visible economic impacts. This indicates the importance of considering both direct and indirect impacts when assessing the full economic consequences of obesity. Furthermore, the imminent availability of precision pharmacotherapies, such as semaglutide, tirzepatide, retatrutide, and a promising pipeline of novel agents, may substantially transform obesity management [67,68]. Current evidence demonstrates that GLP-1 receptor agonists are both effective and cost-effective, not only in promoting weight reduction but also in improving metabolic parameters, thereby mitigating complications associated with obesity [69,70]. As this analysis reflects current standards of care, future access to more effective or more costly treatments could reshape the COI landscape and consequently influence estimates of the obesity-related burden.

Nevertheless, the prominence of GIT diseases as a major source of economic burden (KWD 1.60 billion, $ 5 billion, and 7.80 PPP or 36% of the total) reflects a distinctive aspect of the obesity profile in Kuwait. Probably, the early onset and chronic nature of GIT diseases (NAFL and GERD) in individuals with obesity [71] may lead to persistent healthcare utilization over a lifetime. This burden is further influenced by prevalent dietary habits and sedentary lifestyles among Kuwaitis [72], justifying their prominence in the overall obesity-related burden assessment. At the same time, CVDs, despite a relatively lower number of attributable cases (77,180 morbidity cases) compared to some other conditions, disproportionately exerted the most significant impact in terms of premature mortality (11,576 YPLL). This is likely attributable to the earlier onset of CVDs in individuals with obesity, the large number of deaths related to these diseases in the population, and their elevated case-fatality rates. These findings highlight the critical need for focused cardiovascular risk reduction strategies within obesity management, particularly targeting young and middle-aged adults.

Interestingly, certain conditions—including some cancers, musculoskeletal disorders, and neurological or mental health disorders— display notable morbidity burdens yet relatively modest total costs. This pattern may reflect factors such as later age of onset, and relatively low OAFs for these conditions which may reduce the per capita economic loss. Another possible contributing factor may be underdiagnosis or underreporting, particularly for mental health conditions, which are often under-prioritized in obesity research despite known associations with the disease. Moreover, the disabling nature of these disorders likely results in costs associated more with long-term functional impairment than acute medical care, which in this study were attributed to obesity rather than its comorbidities.

The enormous economic burden of obesity observed in the present study may result from greater obesity prevalence, younger affected populations, scale of prevention efforts, and differences in methodology across studies, such as variations in the inclusion of obesity-related diseases and different cost components. This high economic burden relative to Kuwait's GDP and Ministry of Health budget highlights the potential return on investment in prioritizing obesity prevention and integrated chronic disease management. Of note, Kuwait's national healthcare system is predominantly publicly funded. In the 2023/24 fiscal year, the government allocated approximately 11.5% of its total budget—about KWD 3.04 billion ($10 billion; PPP 15.3 billion)—to healthcare, ensuring free access to public healthcare services for Kuwaiti citizens [57]. Per capita health expenditure was estimated at KWD 520 ($1,700; PPP 2,614) in 2022, with more than 80% of spending directed to the public sector [58].

## Policy implications

In light of these findings, several policy recommendations emerge. First, implement culturally tailored comprehensive public health strategies that address lifestyle factors, dietary habits, social determinants, and clinical care pathways. These strategies may include school-based nutrition programs, public education campaigns promoting traditional Kuwaiti dietary practices, restrictions on marketing unhealthy foods, and taxation of sugar-sweetened beverages. Worthy mentioning, the WHO has updated its list of "best buys" and recommended interventions to combat noncommunicable diseases. Key strategies include community-based public education promoting physical activity, taxation of sugar-sweetened

beverages, and front-of-package nutritional labeling. Additionally, policies that affect the price and availability of fruits and vegetables are advocated to support healthier dietary choices [73]. Second, early detection and management programs for obesity in primary care settings are essential to mitigate progression to high-cost conditions. Third, investments in workplace wellness programs and community-based interventions could reduce productivity losses and promote sustained behavior change. Fourth, the implementation of mobile-based behavior changes tools, AI-driven diet coaching, and wearable-connected monitoring can all play a crucial role in supporting long-term weight maintenance [74,75]. Finally, expanding access to obesity-specific treatments—such as new and innovative pharmacotherapies, behavioral counseling, and metabolic bariatric surgery—should be considered to directly reduce disease incidence and healthcare system strain. This study reinforces the call for serious initiatives to elevate obesity as a public health priority, with particular emphasis on early detection and management across all levels of healthcare facilities.

## Limitations

While this COI study provides valuable insights, certain limitations should be acknowledged. A key limitation of this study is the absence of primary epidemiological and cost data specific to the Kuwaiti context, stemming from the lack of registries tracking multimorbidity and population-level information. To mitigate this, the authors utilized the best available evidence from multiple sources in comparable high-income countries [46], as previously detailed. Strengthening data infrastructure and establishing continuous surveillance systems are crucial to accurately monitor trends, improve cost estimates, and effectively evaluate the cost-effectiveness of various interventions. Although this study employed a broad societal perspective to estimate the economic burden of obesity, it excluded costs associated with overweight among adults and obesity in children and adolescents, representing a notable limitation. Overweight and obesity significantly impact these age groups, suggesting that the total burden is likely underestimated. For instance, a study reported that 40.9% of children aged 6–8 was either had overweight or obesity. Additionally, research on school children aged 5–19 indicated prevalence rates exceeding 50%, which are higher than those observed in other regions and some high-income countries [76]. Future research should incorporate the entire population to provide more comprehensive burden estimates. Similarly, the cost of presenteeism was not estimated in this COI. Moreover, intangible costs such as psychological distress, stigma, and quality-of-life loss were not captured, likely leading to underestimation of the true societal burden.

## Conclusion

Obesity and its associated comorbidities impose a significantly greater health and economic burden in Kuwait than previously recognized, representing a major threat to the healthcare system and national productivity. This underscores the urgent need for comprehensive, population-level preventive interventions. This COI study advocates for a paradigm shift that targets obesity as a root cause by prioritizing early prevention, culturally tailored strategies, and integrated disease management, rather than focusing solely on managing its complications. Expanding access to obesity-specific treatments—including emerging innovative pharmacotherapies—and leveraging technology-driven tools such as mobile behavior change applications, AI-based diet coaching, and wearable monitoring devices are essential to support sustained weight management and alleviate the long-term burden.

Given Kuwait's unique cultural context, public health strategies must be culturally sensitive, incorporating community engagement and gender-specific approaches to effectively address diverse prevalence patterns. Furthermore, bridging existing gaps in epidemiological and economic data through enhanced registry systems and real-world evidence collection is critical to enable accurate monitoring, evaluation, and informed policymaking.

## Supporting information

**S1 Table. Sources of non-cost data.**
(DOCX)

**S2 Table. Annual cost inputs per case (person).**
(DOCX)

**S3 Table. Morbidity cases, mortality cases, costs (KWD), and YPLL attributable to obesity in Kuwait (per disease group).**
(DOCX)

## Acknowledgments

The authors express their sincere gratitude to Mr. Anas Naji for his dedicated efforts in data collection.

**AI Tools Utilization**: During the preparation of this work, the author(s) used Perplexity AI tool to assist with language editing, grammar correction, and to improve overall readability, in accordance with ethical standards.

## Author contributions

**Conceptualization:** Mouaddh Abdulmalik Nagi.

**Data curation:** Mouaddh Abdulmalik Nagi, Hanan Ahmed.

**Formal analysis:** Mouaddh Abdulmalik Nagi, Hanan Ahmed.

**Methodology:** Mouaddh Abdulmalik Nagi, Hanan Ahmed.

**Project administration:** Mouaddh Abdulmalik Nagi.

**Resources:** Mouaddh Abdulmalik Nagi, Hanan Ahmed.

**Supervision:** Mouaddh Abdulmalik Nagi.

**Validation:** Hanan Ahmed, Mohammad Almari, Ziyad S. Almalki, Yasir Mohammed Zaroug Elradi.

**Writing – original draft:** Mouaddh Abdulmalik Nagi, Hanan Ahmed.

**Writing – review & editing:** Mouaddh Abdulmalik Nagi, Hanan Ahmed, Mohammad Almari, Ziyad S. Almalki, Yasir Mohammed Zaroug Elradi.

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
