## [Decision Letter · Decision Letter 0]

21 Jan 2026

A comprehensive economic assessment of the burden of obesity in Kuwait

PLOS One

Dear Dr. Nagi,

Thank you for submitting your manuscript to PLOS ONE. After careful consideration, we feel that it has merit but does not fully meet PLOS ONE’s publication criteria as it currently stands. Therefore, we invite you to submit a revised version of the manuscript that addresses the points raised during the review process.

We look forward to receiving your revised manuscript.

Kind regards,

Humaira Nisar

Academic Editor

PLOS One

Journal Requirements:

2. Please include captions for your Supporting Information files at the end of your manuscript, and update any in-text citations to match accordingly. Please see our Supporting Information guidelines for more information: http://journals.plos.org/plosone/s/supporting-information .

Reviewer's Responses to Questions

**Comments to the Author**

1. Is the manuscript technically sound, and do the data support the conclusions?

Reviewer #1: Yes

2. Has the statistical analysis been performed appropriately and rigorously?

Reviewer #1: Yes

3. Have the authors made all data underlying the findings in their manuscript fully available?

Reviewer #1: Yes

4. Is the manuscript presented in an intelligible fashion and written in standard English?

Reviewer #1: Yes

Reviewer #1: Title and Abstract,

The title accurately reflects the manuscript's focus on assessing the comprehensive economic burden of obesity in Kuwait. The abstract is well-structured, covering the study’s background, methodology, results, and conclusion concisely.

I suggest the following comments to improve the manuscript.

Introduction:

1. Add information about the economic burden (e.g, medical direct cost, non-medical direct costs, and indirect costs) of obesity in Kuwait. Through statistics, the mortality rate and comparison with other countries should be reported. Through statistics, the mortality rate, economic burden, and comparison with other countries should be reported.

2. I invite the authors to write a summary of the importance of COI analysis in health care, and in measuring medical and other costs

3.

Methods:

- To follow this section better, I invite the authors to write this section according to the EQUATOR network and related checklist/guideline.

URL: https://www.equator-network.org/

Discussion

"Policy implication" section is missing (please add before limitation section)

**Do you want your identity to be public for this peer review?** For information about this choice, including consent withdrawal, please see our Privacy Policy

Reviewer #1: No

---

## [Author Response · Author response to Decision Letter 1]

26 Jan 2026

Reviewer #1: Title and Abstract,

The title accurately reflects the manuscript's focus on assessing the comprehensive economic burden of obesity in Kuwait. The abstract is well-structured, covering the study’s background, methodology, results, and conclusion concisely.

I suggest the following comments to improve the manuscript.

Author response:

We wanted to take a moment to show our appreciation for the time and effort you put into reviewing this manuscript. We have addressed all comments, which we agree have further enhanced the paper's clarity and policy relevance.

● Comment: Introduction:

1. Add information about the economic burden (e.g, medical direct cost, non-medical direct costs, and indirect costs) of obesity in Kuwait. Through statistics, the mortality rate and comparison with other countries should be reported. Through statistics, the mortality rate, economic burden, and comparison with other countries should be reported.

○ Response: We thank the reviewer for this very helpful framing suggestion.

○ Action: The text has been revised accordingly. Sentences have been added to provide this context and clarify the economic burden of obesity in Kuwait and the mortality rate and comparison with other countries.

○ Revised Text (Line 93-100): "This analysis used epidemiological and cost data from published studies and global datasets. It estimated Kuwait's 2019 burden at US$2.3 billion—1.7% of the country’s GDP—with projections rising to US$33.3 billion (9.42% of the country’s GDP) by 2060 [3]. In 2019, this compromised direct medical costs (US$638 million), direct non-medical costs (US$360 million), and indirect costs (US$1,660 million). Relative to other countries, Kuwait's obesity-related economic burden ranks among the highest, exceeding many high-income countries (e.g., 1.30% of the national GDP in Korea, 1.53% in Qatar, and 0.96% in Singapore) [3]. Moreover, Kuwait also ranks among the highest countries in terms of mortality attributable to high BMI globally. In 2023, high BMI accounted for 21.77% of all deaths in Kuwait, compared with 8.06% in high-income countries and 6.12% worldwide [14]. Complementing these findings, a retrospective micro-costing study from the public payer perspective focused on ten key obesity-related comorbidities."

● Comment:

2. I invite the authors to write a summary of the importance of COI analysis in health care, and in measuring medical and other costs.

○ Response: We appreciate the reviewer’s insightful comment regarding the importance of COI analysis in health care.

○ Action: We have added a summary as a new paragraph in the Introduction section to address the importance of cost-of-illness (COI) analysis in healthcare and its role in measuring medical and other costs.

○ Revised Text (Line 115-125): " In this context, cost-of-illness (COI) studies offer a comprehensive approach, representing a key form of health economic analysis that quantifies the economic burden of diseases or risk factors like obesity on individuals and society. These analyses delineate direct medical costs (e.g., hospitalizations, pharmaceuticals), direct non-medical costs (e.g., transportation, accommodation), and indirect costs (e.g., productivity losses from morbidity and premature mortality), yielding reliable country-specific estimates vital for evaluating preventive or control interventions [16]. By informing resource allocation, prioritizing policies, raising awareness of obesity's societal impact, and justifying targeted budget measures—particularly in regions where such expenditures rival major health threats—COI studies underpin evidence-based decision-making and strategies extending beyond clinical outcomes [17]."

● Comment: Methods:

3. To follow this section better, I invite the authors to write this section according to the EQUATOR network and related checklist/guideline.

URL: https://www.equator-network.org/

○ Response: We thank the reviewer for this invitation to enhance the Methods section’s structure and alignment with EQUATOR Network guidelines. While no specific reporting guideline exists for cost-of-illness (COI) studies within the EQUATOR Network, our manuscript already adheres to the consensus-based checklist for critical appraisal of COI studies, which promotes methodological rigor, transparency, and international comparability [Schnitzler L, Roberts TE, Jackson LJ, Paulus ATG and Evers SMAA. A consensus-based checklist for the critical appraisal of cost-of-illness (COI) studies. Int J Technol Assess Health Care. 2023;39(1):e34. doi:10.1017/S026646232300019].

○ Action: We have added the following sentence to contextualize the adherence to the consensus-based checklist for critical appraisal of COI studies developed by Schnitzler et al. (2023).

○ Revised Text (Line 148-153): " Importantly, to ensure methodological rigor, reporting quality, and international comparability, this study followed the consensus-based checklist for critical appraisal of COI studies developed by Schnitzler et al. (2023) [21]. This checklist comprises 17 main questions (plus sub questions) across three domains: study characteristics, methodology and cost analysis, and results and reporting. Notably, it has been adopted in several published COI studies [22-23].

● Comment: Discussion

Policy implication section is missing (please add before limitation section)

○ Response: We appreciate the reviewer’s emphasis on the importance of including policy implication section to enhance the relevance and usefulness of this manuscript. A comprehensive policy implication section already exists in the Discussion part, outlining culturally tailored public health strategies, early detection programs, workplace interventions, digital tools, and expanded access to obesity treatments.

○ Action: To enhance its prominence and address this comment, we have added a clear subheading—"Policy implications"—immediately before the Limitations section.

Revised Text (Line: 519): " Policy implications."

---

## [Decision Letter · Decision Letter 1]

11 Feb 2026

Dear Dr. Nagi,

We look forward to receiving your revised manuscript.

Kind regards,

Sreeram V. Ramagopalan

Academic Editor

PLOS One

Journal Requirements:

Reviewers' comments:

Reviewer's Responses to Questions

**Comments to the Author**

Reviewer #1: (No Response)

2. Is the manuscript technically sound, and do the data support the conclusions?

Reviewer #1: Yes

3. Has the statistical analysis been performed appropriately and rigorously?

Reviewer #1: Yes

4. Have the authors made all data underlying the findings in their manuscript fully available?

Reviewer #1: Yes

5. Is the manuscript presented in an intelligible fashion and written in standard English?

Reviewer #1: Yes

Reviewer #1: As a peer reviewer, I have carefully evaluated the submitted manuscript in terms of structure, scientific quality, methodology, and presentation.

Below are my detailed comments and recommendations for each section of the paper.

This paper estimates “The economic costs associated with obesity from governmental and societal perspectives in the State of Kuwait. The subject is of interest and very important in the field of Health Economics.

Abstract:

• This section is enough.

Main text:

Introduction:

1. The introduction section is written well.

Method:

1.Unit costs should be provided.

Result:

1.This section is missing.

Discussion

1. This section is written well. However, I invite the author compare the main results with those conducted in developed countries like the USA.

**Do you want your identity to be public for this peer review?** For information about this choice, including consent withdrawal, please see our Privacy Policy

Reviewer #1: No

---

## [Author Response · Author response to Decision Letter 2]

11 Feb 2026

PONE-D-25-47395R1

A comprehensive economic assessment of the burden of obesity in Kuwait

PLOS One

Dear Dr. Nagi,

Thank you for submitting your manuscript to PLOS ONE. After careful consideration, we feel that it has merit but does not fully meet PLOS ONE’s publication criteria as it currently stands. Therefore, we invite you to submit a revised version of the manuscript that addresses the points raised during the

Date Sent: 11-Feb-2026

Dear Editor,

We are thrilled to have the opportunity to revise our manuscript and would like to express our gratitude to you and the reviewers for providing valuable comments and suggestions. These insights have been incredibly helpful in improving our manuscript. As a result, we have uploaded a revised copy of the manuscript with all the changes made during the revision process highlighted in blue.

We wanted to take a moment to show our appreciation for the time and effort you put into providing such insightful guidance. Our hope is that these revisions will elevate the paper and make it worthy of publication in PLOS ONE Journal. Herein, we offer detailed responses to all the comments and suggestions.

Thank you once again for your help and support.

Sincerely,

Corresponding author

● Comment: Please include the following items when submitting your revised manuscript:

• A letter that responds to each point raised by the academic editor and reviewer(s). You should upload this letter as a separate file labeled 'Response to Reviewers'.

○ Response: We thank the editorial team for these clarifications.

○ Action: The items have been included as instructed.

● Comment: ○ Response: We thank the editorial team for this offer.

○ Action: No action has been taken as we have no changes to our financial disclosure.

● Comment: If applicable, we recommend that you deposit your laboratory protocols in protocols.io to enhance the reproducibility of your results.

○ Response: We thank the editorial team for this recommendation.

○ Action: No action has been taken as this recommendation is not applicable to our manuscript.

Journal Requirements:

● Comment: When submitting your revision, we need you to address these additional requirements.

○ Response: We thank the editorial team for this clarification.

○ Action: No action has been taken as this recommendation is not applicable to our manuscript.

○ Response: We thank the editorial team for this recommendation.

○ Action: We reviewed our reference list and ensured it is complete and correct.

Reviewers' comments:

Reviewer's Responses to Questions

Comments to the Author

1. If the authors have adequately addressed your comments raised in a previous round of review and you feel that this manuscript is now acceptable for publication, you may indicate that here to bypass the “Comments to the Author” section, enter your conflict of interest statement in the “Confidential to Editor” section, and submit your "Accept" recommendation.

Reviewer #1: (No Response)

○ Response: We have no response.

2. Is the manuscript technically sound, and do the data support the conclusions?

Reviewer #1: Yes

○ Response: We wanted to take a moment to show our appreciation for the time and effort you put into reviewing this manuscript.

3. Has the statistical analysis been performed appropriately and rigorously?

Reviewer #1: Yes

○ Response: We wanted to take a moment to show our appreciation for the time and effort you put into reviewing this manuscript.

4. Have the authors made all data underlying the findings in their manuscript fully available?

Reviewer #1: Yes

○ Response: We wanted to take a moment to show our appreciation for the time and effort you put into reviewing this manuscript.

5. Is the manuscript presented in an intelligible fashion and written in standard English?

Reviewer #1: Yes

○ Response: We wanted to take a moment to show our appreciation for the time and effort you put into reviewing this manuscript.

6. Review Comments to the Author

Reviewer #1:

As a peer reviewer, I have carefully evaluated the submitted manuscript in terms of structure, scientific quality, methodology, and presentation.

Below are my detailed comments and recommendations for each section of the paper.

This paper estimates “The economic costs associated with obesity from governmental and societal perspectives in the State of Kuwait. The subject is of interest and very important in the field of Health Economics.

We wanted to take a moment to show our appreciation for the time and effort you put into reviewing this manuscript. We have addressed all comments, which we agree have further enhanced the paper's clarity.

● Comment: Abstract:

This section is enough.

○ Response: We thank the reviewer for this feedback.

○ Action: No action has been taken.

● Comment: Introduction

The introduction section is written well.

○ Response: We appreciate the reviewer’s encouraging statement.

○ Action: No action has been taken.

● Comment: Methods:

Unit costs should be provided.

○ Response: We thank the reviewer for this valuable recommendation. All unit costs are comprehensively detailed in S2 Table (Annual cost inputs per case) within the Supporting Information file, enhancing transparency in the Methods section.

Hope this is satisfying.

● Comment: Results

This section is missing

○ Response: We thank the reviewer for highlighting this issue. Upon submission/resubmission, we confirm that the manuscript includes a comprehensive Results section (lines 329–415, pages 15–21), detailing all findings. This is further supplemented by S3 Table in the Supporting Information file. We apologize for any submission error that may have caused the section to appear missing and have verified its presence in the revised files.

● Comment: Discussion

This section is written well. However, I invite the author compare the main results with those conducted in developed countries like the USA.

○ Response: We thank the reviewer for this constructive suggestion.

○ Action: As suggested, we have incorporated findings from key US studies alongside previously cited high-income country comparisons (Portugal, Saudi Arabia, Italy, Switzerland) to strengthen the international benchmarking in the Discussion section (Line 434-448).

○ Revised Text (Line 444-446): In contrast, earlier estimates reported direct medical costs comprising 43% of the total cost in the USA in 2019 [3] and 23.4% in 2023 [5].

7. PLOS authors have the option to publish the peer review history of their article (what does this mean?). If published, this will include your full peer review and any attached files.

Do you want your identity to be public for this peer review? For information about this choice, including consent withdrawal, please see our Privacy Policy.

Reviewer #1: No

● Comment: To ensure your figures meet our technical requirements, please review our figure guidelines: https://journals.plos.org/plosone/s/figures

○ Response: We thank the editorial team for this recommendation.

○ Action: No action has been taken as this recommendation is not applicable to our manuscript.

---

## [Editor Report · Decision Letter 2]

15 Feb 2026

A comprehensive economic assessment of the burden of obesity in Kuwait

PONE-D-25-47395R2

Dear Dr. Nagi,

We’re pleased to inform you that your manuscript has been judged scientifically suitable for publication and will be formally accepted for publication once it meets all outstanding technical requirements.

Kind regards,

Sreeram V. Ramagopalan

Academic Editor

PLOS One
---

## [Editor Report · Acceptance letter]

PONE-D-25-47395R2

PLOS One

Dear Dr. Nagi,

I'm pleased to inform you that your manuscript has been deemed suitable for publication in PLOS One. Congratulations! Your manuscript is now being handed over to our production team.

Kind regards,

on behalf of

Dr. Sreeram V. Ramagopalan

Academic Editor

PLOS One